# Social networks and risk of delayed hospital arrival after acute stroke

Amar Dhand [1,2], Douglas Luke[3], Catherine Lang[4], Michael Tsiaklides[5], Steven Feske[1] & Jin-Moo Lee[5]

Arriving rapidly to the hospital after a heart attack or stroke is critical for patients to be within time windows for treatment. Prior research in heart attacks has suggested a paradoxical role of the social environment: those who arrive early are surrounded by nonrelatives, while those who arrive late are surrounded by spouses or family members. Here, we used network methods to more deeply examine the influence of social context in stroke. We examined the relationship of personal social networks and arrival time in 175 stroke patients. Our results confirmed the paradox by showing that small and close-knit personal networks of highly familiar contacts, independent of demographic, clinical, and socioeconomic factors, were related to delay. The closed network structure led to constricted information flow in which patients and close confidants, absent outside perspectives, elected to watch-and-wait. Targeting patients with small, close-knit networks may be one strategy to improve response times.

[1] Department of Neurology, Brigham and Women's Hospital, Harvard Medical School, Boston 02115 MA, USA. [2] Network Science Institute, Northeastern University, Boston 02115 MA, USA. [3] Center for Public Health Systems Science, Washington University in St. Louis, St. Louis 63130 MO, USA. [4] Program in Physical Therapy, Washington University School of Medicine, St. Louis 63108 MO, USA. [5] Department of Neurology, Washington University School of Medicine, St. Louis 63110 MO, USA. Correspondence and requests for materials should be addressed to A.D. (email: adhand@bwh.harvard.edu)

Delay in arrival at the hospital is the most important reason for not deploying treatment for heart attacks or strokes in the world. More than 50% of the annual 1.2 million people who suffer a heart attack death in the United States die in the emergency department or before reaching a hospital within an hour of symptoms[1]. Approximately, 70% of the 795,000 stroke patients every year in the United States arrive after 6 h of symptom onset[2]. These delays result in greater death rates and worse functional outcomes after cardiac and neurological emergencies[1,3]. The most common reason for the delay is the time to activate medical care by patients and witnesses who do not recognize the symptoms as serious, do not call an ambulance, or take a watch-and-wait approach[4].

Paradoxically, close family members worsen delay in cardiac emergencies. In a study of 1102 patients with chest pain, the time of evaluation before calling medical care by nonrelatives/coworkers was ~19.7 min, family members ~30.2 min, and spouses ~35.1 min[5]. In other words, the fastest response times were for patients surrounded by coworkers or nonrelatives, whereas family members, particularly spouses, often suggested watch-and-wait approaches. Nonfamily members were "just not as easily negotiated with or dissuaded from usurping control or calling for

medical care" (p. 1308)[5]. In sum, the social situation is a major factor in emergency decision-making and rival the effects of more traditional individual patient factors such as demographic and clinical characteristics.

Examining the social environment in medical emergencies is also appropriate because most heart attacks and strokes occur in the presence of others. For example, 80% of strokes occur in the presence of others and ≈70% occur at patients' homes[6]. Caregivers or witnesses activate ≈96% of stroke emergency calls[7]. Bystanders recognize stroke symptoms more often than patients do, and more frequently advocate for action[8]. Therefore, the earliest moments of a stroke is a group phenomenon involving at least one caregiver and, often, multiple family members, friends, or strangers communicating to determine whether and when action should be taken. Consequently, delay is often the result of the communications, decisions, and actions of multiple witnesses working and thinking together[9,10].

The network perspective provides a set of theories and methods to study the spread of information or behaviors in groups. Research has shown that exercise[11], weight gain[12], and medication use[13] may be contagious behaviors that flow through ties to shape and constrain personal choices. A formal way to study these influences is through well-known sociological theories known as the strength of weak ties[14] and structural holes[15]. These approaches illustrate two archetypal network formations of social capital: dense personal networks ideal for social support (e.g., bonding capital) and radial personal networks optimal for access to novel information (e.g., bridging capital)[16]. Through these structural motifs, personal networks act as conduits of health-related social capital to identify symptoms, recognize a need for support, and help secure access to services[17], all of which are critical in medical emergencies.

The purpose of this study is to understand the arrival to the hospital as a collective process using network methods. We use personal network analysis, also known as egocentric network analysis, to map and understand the influence of the persons who surround a stroke patient[18]. Specifically, we analyze patients' personal network characteristics such as size and connectivity, which represent conduits for information entry, exchange, and disruption. We also examine the characteristics of each network member to understand the diversity or likeness of potential witnesses. Finally, we analyze descriptions of the communication patterns that led to early or delayed arrival with attention to underlying social mechanisms. The findings expose the unexpected role of patients' social networks to shape decisions during medical emergencies.

## Results

**Generating personal network data in stroke patients.** We used a survey to characterize the personal social network data of 175 patients within 5 days of stroke onset. The survey used traditional name generators from General Social Survey[19,20] to create network maps. For each patient, we were particularly interested in quantifying the opportunities for the patient to receive and control new information. Drawing on Burt's concept of social capital, we quantified the holes in the social structure that allow the patient to be exposed to different flows of information. For example, in Fig. 1, Patient A has a small and close-knit, or high constraint, network with no structural holes because each network member is strongly connected to all the others leading to restricted information flow. Patient B has a large and radial, or low constraint, network with structural holes as the subgroup of friends (left) is not known to the other subgroup (right) leading to two separate flows of information. These principles were operationalized by calculating network

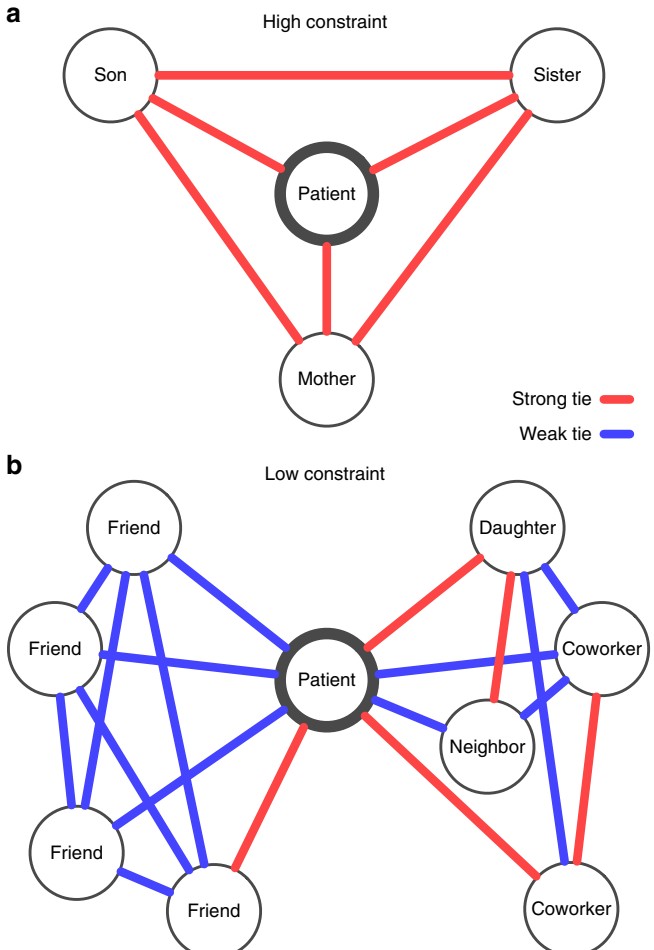

**Fig. 1** Personal networks of two patients with ischemic stroke. Each patient is connected to social contacts by red lines indicating strong or especially close relationships and blue lines indicating weak or less close relationships. No ties between persons means those individuals do not know each other. Patient A has a high-constraint network and Patient B has a low-constraint network

**Table 1 Baseline characteristics of patients, according to arrival time**

| Characteristic | Slow arrivers (n = 67) | Fast arrivers (n = 108) | P value[a] |
|---|---|---|---|
| Age, mean (sd), y | 61.0 (15.9) | 61.7 (15.6) | 0.7717 |
| Female sex, No. (%) | 32 (48) | 58 (54) | 0.5425 |
| Race, No. (%)[b] | | | |
| White | 37 (55) | 78 (72) | 0.0309 |
| Black/African American | 28 (42) | 27 (25) | |
| Other | 2 (3) | 3 (3) | |
| Years of education, median [IQR] | 14 [12, 15] | 14 [12, 16] | 0.2136 |
| Median household income [IQR], $ thousand | 39.0 [32.3, 54.8] | 48.9 [35.0, 59.5] | 0.0518 |
| Insurance, No. (%) | | | 0.7039 |
| Private | 34 (51) | 59 (55) | |
| Medicare | 12 (18) | 12 (11) | |
| Medicaid | 2 (3) | 6 (6) | |
| Uninsured | 17 (25) | 27 (25) | |
| Veteran | 2 (3) | 4 (4) | |
| Married, No. (%) | 24 (36) | 52 (48) | 0.1492 |
| Living alone, No. (%) | 22 (33) | 27 (25) | 0.3426 |
| NIHSS score, mean (range)[c] | 3 (0–12) | 3 (0–13) | 0.4229 |
| Ischemic stroke severity, No. (%) | | | |
| Mild (NIHSS < 6) | 61 (91) | 93 (86) | 0.4229 |
| Moderate (NIHSS 6-15) | 6 (9) | 15 (14) | |
| Severe (NIHSS > 15) | 0 (0) | 0 (0) | |
| Side of stroke, No. (%)[d] | | | |
| Left | 16 (36) | 23 (30) | 0.7267 |
| Right | 27 (61) | 50 (66) | |
| Both | 1 (2) | 3 (4) | |
| Area of stroke, No. (%) | | | |
| Cortical | 11 (16) | 27 (25) | 0.4319 |
| Subcortical | 32 (48) | 45 (42) | |
| Both cortical and subcortical | 1 (1) | 4 (4) | |
| Brainstem or cerebellum | 23 (34) | 32 (30) | |
| Aphasia, No. (%) | 3 (4) | 3 (3) | 0.8624 |
| Arrived by EMS, No. (%) | 37 (55) | 74 (69) | 0.1066 |
| Charlson comorbidity index, median [IQR][e] | 2 [1, 4] | 3 [1, 4] | 0.2502 |
| Short-blessed test score, median [IQR][f] | 2 [0, 4] | 2 [0, 4] | 0.2010 |
| Patient health questionnaire-9, median [IQR][g] | 3 [1, 6.5] | 2 [0, 5] | 0.1111 |
| Tissue plasminogen activator given, No. (%)[h] | 0 (0.0) | 54 (50) | <0.0001 |

[a] P value calculated from unpaired two-tailed t test or Wilcoxon rank sum test for continuous variables and $\chi^2$ test and Fisher's exact test for categorical variables
[b] Statistical test run on Black/African American race versus not Black/African American
[c] NIHSS is the National Institutes of Health Stroke Scale, a measure of stroke severity ranging from 0 (mild) to 42 (severe)
[d] Side of stroke determination did not include brainstem strokes
[e] Charlson Comorbidity Index is a weighted index to predict risk of death with specific comorbid conditions
[f] Short-blessed test score is an orientation-memory concentration test of cognitive impairment
[g] Patient health questionnaire-9 is a screening test for depression
[h] Tissue plasminogen activator is a medication to dissolve clots usually given within 4.5 h of stroke

size (number of nodes in network), Burt's constraint (a measure of close-knit structure), effective size (an inverse of constraint with higher values indicating radial structure), and mean degree (average number of ties per node) (see Methods for mathematical descriptions).

We recruited a diverse set of patients with mostly mild motor-predominant stroke (88% mild, 12% moderate). We focused on mild stroke because patients with milder symptoms are at higher risk of delay, and they were able to engage in the survey during hospitalization. Table 1 lists their baseline demographic and clinical characteristics. Of the 175 study participants, 108 arrived at the hospital ≤6 h, and 67 arrived >6h after symptom onset. We found that demographic and clinical traits did not differ between individuals who refused and those who agreed to participate in the study. We did note that delayed arrivers were more likely to be Black/African American (42 vs. 25%, Fisher's exact test $p = 0.0309$), and were less likely to be given tissue plasminogen activator, a medication that dissolves blood clots given only if stroke patients arrive within 4.5 h (0% vs. 50%, two-tailed t test $p < 0.0001$). The difference in treatment frequencies highlights one of the main consequences of arriving late.

**Small close-knit personal networks and slow arrival**. The networks of slow arrivers were smaller and more close-knit than the networks of fast arrivers. This was first evident when we created a montage of all patients' networks that showed slow arrivers' networks were smaller and less spiky than fast arrivers' networks (Fig. 2). In statistical analysis of Burt's social capital measures (Table 2), unadjusted analysis confirmed significant differences in network structure. Slow arrivers had smaller networks (mean 5, interquartile range (IQR): 4–8) than fast arrivers (mean 8, IQR: 6–10, Wilcoxin rank sum test $p < 0.0001$). They also had higher constraint (mean 61.11, IQR: 39.55–66.94) than fast arrivers (mean 40.31, IQR: 35.85–54.09, Wilcoxin rank sum test $p < 0.0001$), meaning a more close-knit redundant network structure with constricted information flow. Slow arrivers also had smaller effective size and mean degree of network members, which further supported the finding.

Analysis of network composition showed less-pronounced differences. Slow arrivers had networks with a smaller range of member ages (measured by the standard deviation of ages) and a higher percentage of nonexercisers. The two groups did not differ in the percentage kin, diversity of sex, or diversity of race.

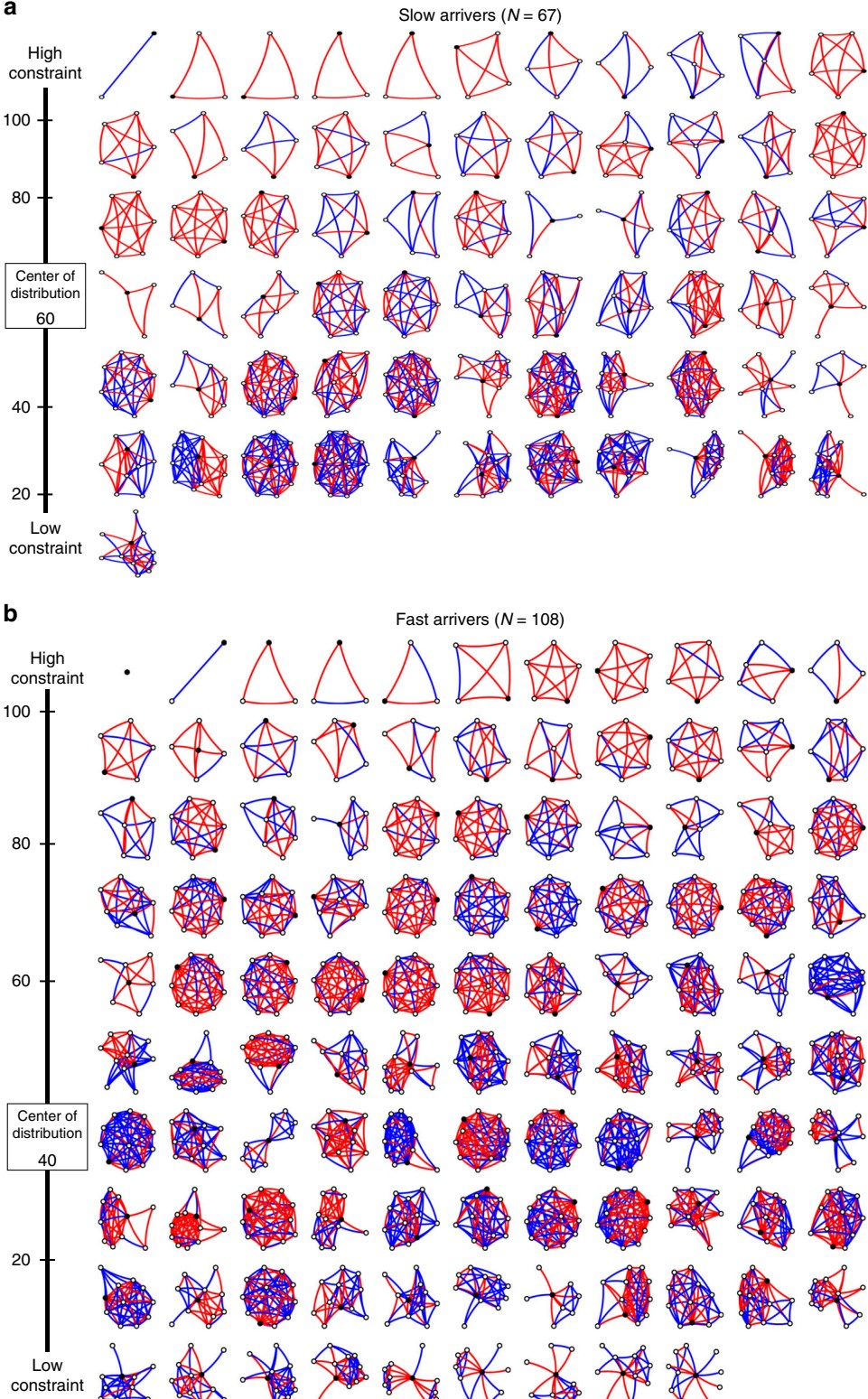

**Fig. 2** Personal networks of patients with ischemic stroke, according to slow and fast arrival. Each black circle represents one patient in the study embedded inside his/her social network. Line color indicates strength of relationship: red lines are strong ties and blue lines are weak ties. Networks are arranged from highest constraint (top left) to lowest constraint (bottom right) with scale and median along the left margin. In general, slow arrivers' networks had higher constraint compared to fast arrivers' networks

**Table 2 Network characteristics of slow and fast arrivers, unadjusted**

| Variable | Slow arrivers (n = 67) | Fast arrivers (n = 108) | P value[a] |
|---|---|---|---|
| Network structure characteristics[b] (Median [IQR]) | | | |
| Network size | 5.00 [4.00-8.00] | 8.00 [6.00-11.00] | <0.0001 |
| Constraint | 61.11 [39.55-66.94] | 40.31 [35.85-54.09] | <0.0001 |
| Effective size | 2.50 [1.86-3.61] | 3.17 [2.21-5.00] | 0.0021 |
| Mean degree | 3.20 [2.00-5.00] | 5.00 [3.11-6.92] | 0.0003 |
| Network composition characteristics[c] (Median [IQR]) | | | |
| Percentage kin | 62.50 [33.33-100] | 57.14 [33.33-6.39] | 0.3285 |
| Standard deviation of ages | 12.01 [6.18-17.28] | 13.43 [10.48-16.82] | 0.0407 |
| Diversity of sex | 0.89 [0.64-0.99] | 0.89 [0.75-0.96] | 0.7611 |
| Diversity of race | 0.00 [0.00-0.00] | 0.00 [0.00-0.00] | 0.6389 |
| Percentage nonexercisers | 62.50 [31.67-80.00] | 44.44 [20.00-66.67] | 0.0125 |

[a] P value calculated from Wilcoxon rank sum test, and not adjusted for other factors
[b] Network structure is a quantitative description of the arrangement of social ties in each patient's personal network. See definitions of each term in Methods
[c] Network composition is the range of characteristics of people around the patient. See definitions of each term in Methods

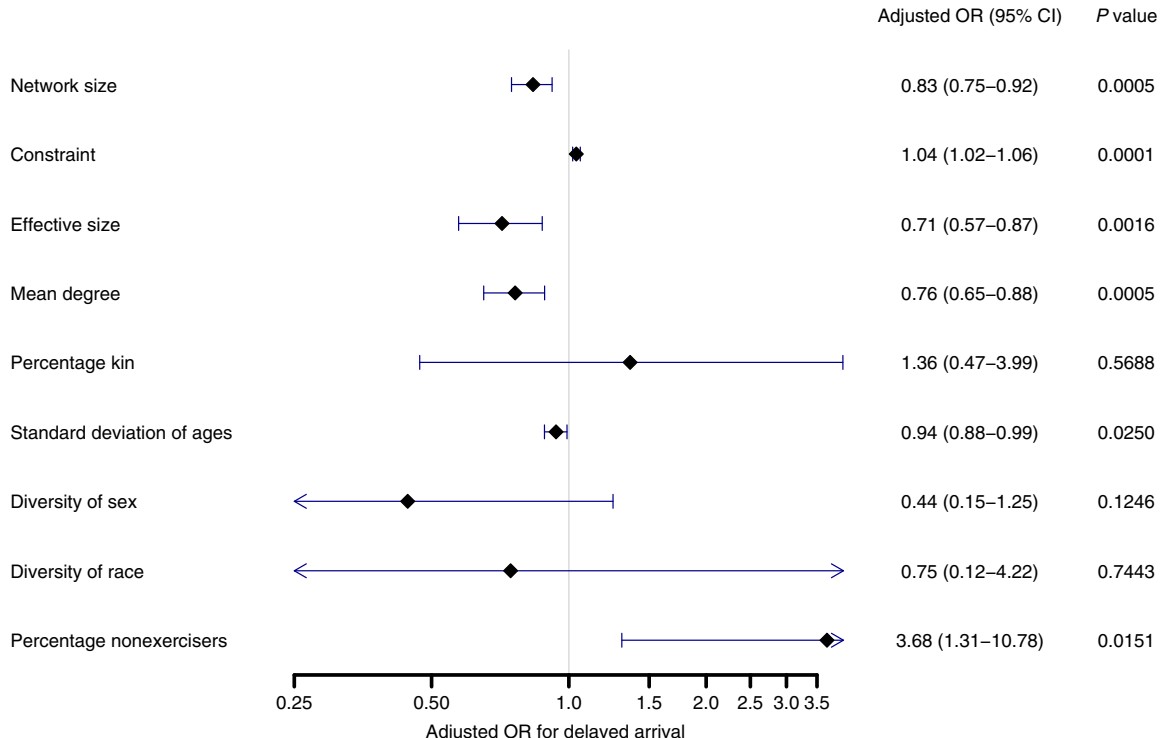

Fig. 3 Forest plot of network metrics and risk of delayed arrival (>6 h). Each diamond is the odds ratio (OR) and each line is the 95% confidence interval (95% CI) of a network metric after adjustment for covariates. P values calculated from multivariable logistic regression. Network structure characteristics (first four rows), more than network composition characteristics, were associated with delayed arrival

The results were similar after adjusting for known confounders (Fig. 3). Based on the literature and our own univariate analyses, we adjusted for age, stroke severity, emergency medical services usage, living alone, education, race, and median income. Network size and constraint remained strongly associated with increased odds of delayed arrival in a multivariable analysis. Network size had an adjusted odds ratio of 0.83 (95% confidence interval [CI]: 0.75–0.92, $p = 0.0005$), inversely related to delay. Constraint had an adjusted odds ratio of 1.04 (95% CI: 1.02–1.06, $p = 0.0001$), positively related to delay. Again, this was supported with the other structural metrics that followed the pattern of closed network structures relating to delay and open network structures relating to early arrival. Thus, effective size and mean degree were strongly associated with lower odds of delayed arrival. The

network composition measures were less markedly related to arrival time, though remained significant after adjustment. Range of the ages of network members (measured by the standard deviation of ages) was associated with lower odds of delayed arrival. Percentage of nonexercisers was associated with higher odds of delayed arrival.

**Constricted information flow in small close-knit networks**. We completed semistructured interviews in a subgroup of patients and used qualitative analysis to understand the communication sequences leading to hospital arrival. In 12 slow and 15 fast arrivers, we found differences in the communication partners and content regarding symptoms and plans (Supplementary Table 1).

In 75% (9 persons) of slow arrivers, the first person contacted or present at time of stroke was an emotionally close, strongly tied person, such as a family member. Slow arrivers' discussions with their strongly connected confidants followed a spiraling pattern toward nonaction: (1) The patient disclosed symptoms in a delayed or selective manner. For example, one person stated, "I wanted to wait and see what happens first." (2) Symptoms were over-negotiated. For instance, in a lengthy argument, one patient declared to her husband, "No, you just wait. I'll just go see my doctor in the morning." (3) Patients and caregivers reinforced each other's opinions to watch and wait. For example, a patient's husband and sister agreed and convinced the patient that the left-sided numbness was not "anything to be concerned about… because…it would come and just last for a few minutes and then go away."

In contrast, only 47% (7 persons) of fast arrivers contacted or were with a strongly tied person, and a weak tie such as a friend or stranger was more often involved. The content of discussion also differed across the groups. Fast arrivers disclosed symptoms quickly, did not negotiate, and did not validate others' plans. For instance, coworkers, who were weak ties in one patient's network, called 911 without discussion with the patient and said, "Something is wrong with you. You need to go to the doctor." These more efficient communication sequences involved more distant contacts leading to faster responses. In Supplementary Movie 1, one patient describes two separate episodes of neurological dysfunction in different social contexts associated with divergent arrival times. Overall, the communication patterns supported the idea that weak ties, which are more available in the large and radial networks, were more likely to result in timely arrival at the hospital.

**Sensitivity analyses to verify findings**. We completed sensitivity analyses to examine the results with a time cutoff of 3 h, removal of very small networks, and stratification by race. The associations of network structure (e.g., constraint) and composition variables (e.g., percent kin) with delayed arrival did not change with a time cutoff of 3 h (Supplementary Fig. 1) or with removal of very small networks (Supplementary Fig. 2). Among Black/African American participants ($n = 57$), associations of network structure and composition with delayed arrival were not significant, although the small sample size limited interpretation (Supplementary Fig. 3). In persons who were not Black/African American ($n = 124$), network structure variables were significant, but network composition variables were not (Supplementary Fig. 4). There was no significant interaction between Black/African American race and the strongest network variables: constraint (adjusted OR = 0.98, 95% CI: 0.95–1.02, $p = 0.39$) or network size (adjusted OR = 1.11, 95% CI: 0.89–1.35, $p = 0.33$). Therefore, we concluded that the findings were robust when considering a shorter time window and not driven solely by the presence of very small networks. We also concluded that the strongest network effects were not moderated or modified by race.

## Discussion

Our results confirm the paradoxical role of the social environment in medical emergencies. We found that stroke patients with small and close-knit social networks had delayed hospital arrival because of restricted information flow that reinforced the norm to watch-and-wait. Conversely, patients with large and radial networks arrived sooner because of access to more diverse information that led to more rapid recognition and action. Networks that included persons with a narrow range of ages or who did not exercise were also risk factors for delayed arrival, although less

markedly. These associations were independent of typical risk factors, such as age, stroke severity, emergency medical services usage, living alone, socioeconomic status (education and median income), and race.

The findings suggest a general solution to the social environmental paradox in heart attacks and strokes. Family member cause a patient to delay not because of their particular blindness to the problem. Rather, the patient is likely embedded in a closed social network structure in which family members are more prevalent. Such a social structure provides reinforcement for prevailing norms due to reduced access to information or diversity of opinions. This results in a communication sequence in which symptoms are not disclosed promptly, plans are over-negotiated, and strategies to wait are unproblematically validated. Such an interpretation is supported by the literature on the importance of the broader social context on risk of delay in multiple countries[9,10,21,22]. Our study moves beyond current literature by using network mapping to reveal social processes rather than proxies of social life such as living alone, being unmarried, or summative social network indices[9]. Our analysis suggests that social capital, defined by network metrics, serves a critical role in medical emergencies.

The importance of weak ties and the potential for misperceptions in closed networks have strong theoretical and empirical support in sociology[14,15]. Weak interpersonal ties have been shown to provide diverse ideas and disruptive opinions informed by unique experiences not shared by other members of the network[23]. For instance, individuals in networks rich with weak ties benefit by acquiring information and brokering diverse resources that lead to success in getting a job, being promoted, or adjusting to new circumstances[24]. Conversely, in strongly tied, close-knit groups, information is recycled inside an "echo chamber," and all persons overestimate the collective support for existing norms. This leads to selective disclosure (e.g., not revealing symptoms immediately) and false consensus-building (e.g., agreeing to watch and wait), both of which were evident in our data[25]. Some researchers have called this phenomenon "pluralistic ignorance"[25] or the "majority illusion," in which an individuals' local observations are distorted by the choices and behaviors of their closest social contacts.

Increased focus on social context could benefit stroke preparedness campaigns in multiple ways. First, our findings suggest a shift from mass education of stroke symptom recognition, which has not substantially improved arrival times[2,26], to a more socially targeted approach. We suggest that practitioners should use a brief survey to map patients' networks during a primary care visit or hospitalization for transient ischemic attack, stroke, or myocardial infarction and identify patients who have small and close-knit networks. At-risk patients and available network members could be offered a network intervention that uses social network data to enhance behavior change. For example, the at-risk patient and network may be taught about stroke symptoms, communication pitfalls, and development of an action plan that avoids pitfalls, such as calling a designated friend. Further study is needed on the efficacy, implementation, and cost-effectiveness of such interventions.

The strengths of our study include the use of social network analysis and mixed methods to understand the social contexts of patients with acute stroke. We also controlled for known demographic, clinical, and socioeconomic factors involved in timely hospital arrival. Finally, we offered a mechanistic understanding of the social influences supported by clinical and sociological literature that suggests ideas for new interventions.

Limitations of our study include the effect of unmeasured confounders. Although we addressed this issue in several ways including measuring factors described in the literature and using

a multivariable regression, it is still possible that unmeasured confounders could explain the observed difference in arrival time. In addition, our results have limited generalizability. Our cohort included patients with mostly mild stroke. Given that mild stroke is a risk factor for delayed arrival[4], we were examining a high-risk cohort. However, since severity predicts time to arrival, social networks may not be important in moderate or severe stroke. Moreover, selection bias may have been introduced by a small sample size of slow arrivers and exclusion of patients who could not participate in the interview survey, such non-English speakers, patients with severe strokes or aphasia, and patients who left before recruitment. Study of a more diverse population is warranted. We also did not assess knowledge of stroke symptoms in the participants, which could have been a covariate or mechanism that deserves further study. Lastly, the survey relied on patients' self-reports of their social networks. Although participants' report of their personal networks of intimate contacts has been shown to be accurate[27], there is still a possibility that patients had recall bias of their social context influenced by their journey to the hospital. A future direction of research could be to survey the network members after stroke or heart attack to understand why and how relatives choose a watch and wait approach more than nonrelatives.

In summary, the decision to seek medical care after stroke is often a social process. Our study shows that patients with small and close-knit social networks are more likely to have delayed hospital arrival after stroke. Our communication analyses suggest this is because slow arrivers tend to contact strongly tied persons with whom they selectively disclose symptoms, over-negotiate plans, and reinforce a watch-and-wait strategy. This is in contrast to fast arrivers who are more likely to contact or be surrounded by weakly tied individuals who act without negotiation. Network interventions that identify patients with small and close-knit networks before stroke and offer strategies to avoid communication pitfalls deserve further study to improve stroke preparedness.

## Methods

**Study design and participants**. The study was a cross-sectional design in which we recruited patients with mostly mild stroke from two academic hospitals, Barnes Jewish Hospital in St. Louis, MO and Brigham and Women's Hospital in Boston, MA, between May 2014 and May 2017. The primary outcome was binary in the form of arrival ≤6 h or >6 h of stroke symptom onset based on medical records, review of acute treatments administered, and patient report. Stroke symptom onset was defined as the time a patient became aware of symptoms. We also did a sensitivity analysis for arrival before or after 3 h. All arrival timings were adjudicated by at least two study team members, one of whom was a neurologist (A.D.). We chose the 6 h threshold because it has been associated with favorable clinical outcomes, irrespective of whether patients receive reperfusion therapy or not[3]. Particularly in mild-stroke patients, who may not receive thrombolysis, this cutoff is associated with favorable outcomes due to a combination of early therapy, monitoring, and intervention if patients worsen[3]. We chose a categorical outcome, rather than continuous, because we could not determine the times with enough precision or accuracy for a meaningful continuous outcome.

We enrolled consecutive patients during their hospitalization if they were (1) diagnosed with a first ischemic stroke, (2) 21 years and older, and (3) within 7 days of clinical stroke. Patients were excluded if they had any of the following: (1) prior ischemic or hemorrhagic stroke, (2) National Institutes of Health Stroke Scale (NIHSS) score > 21, (3) significant aphasia (score > 1 on the language section of the NIHSS), (4) inability to speak English, (5) lack of capacity to consent or participate in the survey interview, (6) diagnosis of dementia or short-blessed test score > 6.

All participants provided written informed consent. The institutional review boards at Barnes Jewish Hospital and Brigham and Women's Hospital approved all study protocols and consent forms. For the movie, the participant provided consent to publish her image.

**Measurements**. A study coordinator administered the social network survey at the patient's bedside, usually between days 2 and 5 of stroke hospitalization. The instrument was an adaptation of the General Social Survey[20] and a national survey of personal networks and health[28]. The main sections were a name generator, name inter-relater, and name interpreter. In the name generator section, participants

named people with whom they had discussed important matters, socialized, or sought support in the last 3 months. In the name inter-relater section, participants determined the connections among all persons in the network and evaluated the strength of the relationship ties. In the name interpreter section, participants answered questions about characteristics and health habits of each individual in the network. Specific question forms are provided in Supplementary Methods 1[18], and instrument psychometric properties are described by Burt[20].

Network structure is a quantitative description of the arrangement of social ties in a patient's social network. Network size is defined as the number of individuals in the network, excluding the patient. Constraint is the degree to which each network member is connected to the others. Constraint summarizes network size (larger networks are less constraining), density (networks with more connectivity are more constraining), tie strength (networks with more strong ties are more constraining), and hierarchy of ties (networks in which all contacts are exclusively tied to a single contact are more constraining)[24,25]. Effective size is the number of nonredundant members in the network, conceptually an inverse metric of constraint[29]. Notably, although we recorded the first responder only for the subgroup who provided qualitative data (discussed below), constraint and effective size account for the number of weak vs. strong ties available to the patient during the emergency. Therefore, these are indirect measures of who could be present at the time of stroke. Mean degree is the average number of ties of a network member, excluding the patient, indicating the distribution of ties in the network. Equations to calculate these measures are provided in the Supplementary Methods 2.

Network composition is the proportion of characteristics across members in the network. Percentage kin is ratio of network members who are family. The standard deviation of network members' age reflects the range of ages of people in the network around the patient. The diversity of sex index (or the index of qualitative variation) represents the mix of men and women in the network with a value of 0 meaning all network members are one sex and a value of 1 indicating equal mix of men and women[30]. The diversity of race is the mix of races in the network with a value of 0 indicating that all persons were of the same race. The percentage of network members who do not exercise or had a stroke are the proportion of individuals who fit each of those categories.

**Statistical analysis**. We compared demographic and clinical characteristics between fast and slow arrivers using an unpaired two-tailed $t$ test or Wilcoxon rank sum test (if nonparametric distribution) for continuous variables and $\chi^2$ test or Fisher's exact test (if less than five in a cell) for categorical variables. We then completed a series of univariate logistic regressions of the demographic, clinical, and network characteristics in relation to arrival time before or after 6 h.

Multivariable analysis was used to determine the association between the strongest social network variables in univariate analysis and delayed arrival. We adjusted for age, stroke severity, emergency medical services usage, living alone, education, race, and median income. The first four variables are well described in the literature as risk factors for delayed hospital arrival[4,9]. We used the last three to determine whether any social network effects are independent of socioeconomic status[26].

We performed a series of logistic regression models using sequential block-wise entry. We tested each network metric separately, as advised in the network literature, due to collinearity[31]. All hypothesis tests were two-sided with the level of significance set at $P$ value < 0.05. The sequential models allowed assessment of individual covariates and comparison of model fit using the log likelihood.

We performed a number of sensitivity analyses including testing our results with a different time cutoff (less than or greater than 3 h), and whether removal of the very small networks (fewer than four members) would alter the results. We also examined the effects of race by stratified and interaction analyses of Black/African American race. All analyses were completed in R version 3.3.2[32].

**Qualitative analysis**. Using semistructured interviews with a subgroup of patients, we asked about the journey to the hospital, including recognition of symptoms, decision-making, and context of the stroke onset. Open-ended questions, included (1) What were the steps you went through from the start of stroke symptoms to arriving at the hospital? (2) Who did you first speak to about your symptoms? (3) Why did you choose to speak with that person? (4) Were you in a public place? (5) Who was the first person to help you? (6) How did you decide how to get to the hospital? (7) What was the most important factor in getting to the hospital? Interviews were audio-recorded and transcribed.

We used qualitative analysis to understand the arrival process and relate it to quantitative patterns. Using the framework method[33], we read and reflected on all interviews, coded recurrent ideas, grouped ideas into themes, and formalized these into a framework and coding index. Finally, we applied the agreed upon index to all transcripts and examined the relationship of the qualitative themes with the quantitative patterns. Two investigators (A.D. and M.T.) independently coded transcripts, agreed on the coding index, and discussed and resolved all disagreements in the final coding.

## Code availability

An updated version of the instrument called "Personal Network Survey for Clinical Research" is available in the REDCap Shared Library. We have also uploaded a

comprehensive R codebase for researchers who use the instrument to analyze and visualize their data available at: https://github.com/AmarDhand/PersonalNetworks. R code used specifically for this project can be made available upon request.

## Data availability
The data used in this study is freely available as a supplement to this manuscript (Source Data).

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

## Acknowledgments
We were supported by National Center for Medical Rehabilitation Research (K23HD083489, PI: Dhand), American Heart Association (14CRP20080001, PI: Dhand), National Institute of Diabetes and Digestive and Kidney Diseases (P30DK046200, PI: Corkey), National Institute of Neurological Disorders and Stroke (R01NS085419, PI: Lee), and the Football Players Health Study at Harvard University. The Football Players Health Study is funded by a grant from the National Football League Players Association. The content is solely the responsibility of the authors and does not necessarily represent the official views of Harvard Medical School, Harvard University or its affiliated academic health care centers, the National Football League Players Association, or Brigham and Women's Hospital. We also wish to acknowledge Angela H. Kim, Karen Li, Abby Halm, and Liam McCafferty for aiding in manuscript preparation.

## Author contributions
A.D., D.L, C.L. and J.L. conceived the study. A.D. and M.T. collected the data. A.D., D.L, C.L., M.T., S.F. and J.L. performed the data analysis. A.D., D.L, C.L., M.T., S.F. and J.L. wrote the paper.

## Additional information

**Competing interests:** The authors declare no competing interests.

