## [Peer Review File · Nature Communications]

Reviewers' comments:

Reviewer #1 (Remarks to the Author):

This is an interesting paper presenting novel data on the relationship between social network structure and time to presentation after the development of stroke symptoms. The authors refer to similar work done with patients suffering from cardiac emergencies and the results of this paper parallel those results. Patients with tight closed networks are more likely to have delayed presentations than those with looser networks.

The data will be of interest to clinicians, social scientists and policy people working to try and increase access to stroke treatments to populations. I have few major comments to make.

1. I don't understand why the paper is laid out in the way it is with the methods presented after the results. It maybe that this is the house style for nature Communications although I couldn't see this in the instructions to authors. I found it more difficult to understand the results not having first read the methods

2. A number of key factors determining speed of response to stroke have been identified in the past. These include knowledge of stroke symptoms and understanding that stroke is a treatable disease. It would have been helpful to have assessed these specific areas of knowledge to ascertain whether there was a systematic difference between the fast and slow responders. These factors are associated with level of education and while the paper shows no difference between years of education there was a difference in household income (nearly significant) which may be another marker of intellect and or personality.

3. I couldn't see reference to who was with the patient at the time of stroke which again is probably a predictor of speed of response and should be considered when analysing the difference between the two groups

4. I did not find the network maps in Figure 2 to be particularly informative – it maybe best if this was in the supplementary information rather than the main text

Reviewer #2 (Remarks to the Author):

Review for:

“Social Networks and Risk of Delayed Hospital Arrival after Acute Stroke”

Dhand et al. 2018

In this paper, the authors present a study that examines the relation between ego-network structure and delayed arrival at the hospital of 175 mild-stroke patients. The results show that small and close personal networks with strong ties -- after controlling for several confounders -- they are related with late arrival at the hospital.

There are definitely several limitations of the current study (including many that the authors acknowledged in the discussion). I would summarize the limitations as the inability to be confident that the study identifies causal evidence. On the other hand, the paper is very well written, the study technically sounds, and the topic is very interesting in both clinical and social research. For those reasons, I believe an improved version could be accepted for publication in Nature Communications. I have some remarks that need to be addressed:

1. I believe, if we exclude all other confounders, the fact that the authors do not consider as control if the patient was surrounded by “at least one ego-network member” at the time of the stroke is problematic. Do the authors have this information?
2. I also believed that the 6-hour threshold was a bit arbitrary, until the authors addressed this at the robustness check section. My question is, why the authors did not consider the 4.5h threshold that a patient can get the tissue plasminogen activator.
3. The authors do not explain why they choose to work with a logistic regression. Why they do not consider the outcome to be just the continuous variable of delay time?
4. Table 2 results are little bit useless, since they are just unadjusted p-values of differences. I would suggest moving it to the supplementary materials – but I am not pushing on that comment.
5. In the introduction there is no reference to networks effects, social capital/Burt’s concept, structural diversity/holes etc. There is a paragraph in the discussion, but I would suggest introducing these concepts in advance – especially for the audience that is not related to network science. Therefore, I am suggesting spending some lines on network effects (e.g. social influence in health behaviors – Christakis & Fowler, Strully et al., Aral & Nicolaides etc.) and then introduce the Burt’s concept (Burt 2000) and strong and weak ties and structural diversity (Granovetter and thereafter work). As it is written, the authors refer directly in the results such concepts and may be difficult for the readers to follow.
6. Also, I would suggest If the authors want to understand the reasons why closed relatives prefer a ‘watch and wait’ pattern in relation to non-relatives, they could conduct surveys to the network of the patients rather than the patients themselves.

Reviewer #3 (Remarks to the Author):

The authors report that a small, closed social network structure is an independent predictor of delayed arrival after stroke onset. The study represents a fascinating new window into delayed arrival, the most important barrier to disease-changing stroke treatment, and will require acknowledgment in future studies of the topic. Strengths include clear expression of sociological concepts, sophisticated statistical analysis and inclusion of a semi-structured interview of a subgroup of patients to explore decision-making at stroke onset. However, the small study population (n=175) accumulated in two academic hospitals over 3 years, differed from the general stroke population in several important ways, some of which may be related to delayed arrival. It excluded patients who did not speak English or otherwise lacked capacity to participate in the survey interview, resulting in a very low average NIHSS of 3. Although low NIHSS is a powerful predictor of late arrival, only 38% arrived more than 6 hours after onset and 31% received IV tPA, a much higher rate than seen in the overall stroke population. Furthermore, the population was very young (average age 61) and highly educated (average 14 years of schooling). The authors appropriately identify the non-representative nature of the cohort as an important limitation but highlight exclusion of severe strokes when the NIHSS cutoff for the study was 19, far higher than the average. Incapacity to participate in the questionnaire was likely a more important exclusion and might be highlighted instead. More broadly, would the authors consider characterizing the study throughout the paper as one of "mild stroke," not just "stroke." The impact of social networks likely diminishes when symptoms are more severe and this study has no capacity to assess that relationship.

Social Networks and Risk of Delayed Hospital Arrival after Acute Stroke

Response to the Referees' Comments

Reviewer #1 comments:

This is an interesting paper presenting novel data on the relationship between social network structure and time to presentation after the development of stroke symptoms. The authors refer to similar work done with patients suffering from cardiac emergencies and the results of this paper parallel those results. Patients with tight closed networks are more likely to have delayed presentations than those with looser networks.

The data will be of interest to clinicians, social scientists and policy people working to try and increase access to stroke treatments to populations. I have few major comments to make.

- 1. I don't understand why the paper is laid out in the way it is with the methods presented after the results. It maybe that this is the house style for nature Communications although I couldn't see this in the instructions to authors. I found it more difficult to understand the results not having first read the methods.**

It is our understanding that this is the house style of *Nature Communications*. We are happy to move based on the Editor's preference.

- 2. A number of key factors determining speed of response to stroke have been identified in the past. These include knowledge of stroke symptoms and understanding that stroke is a treatable disease. It would have been helpful to have assessed these specific areas of knowledge to ascertain whether there was a systematic difference between the fast and slow responders. These factors are associated with level of education and while the paper shows no difference between years of education there was a difference in household income (nearly significant) which may be another marker of intellect and or personality.**

We do not have these data. We have included this as a limitation.

P.12: "We also did not assess knowledge of stroke symptoms in the participants, which could have been a covariate or mechanism that deserves further study."

- 3. I couldn't see reference to who was with the patient at the time of stroke which again is probably a predictor of speed of response and should be considered when analysing the difference between the two groups.**

Although we do not have data on the first responder for all patients, it is captured in the subgroup of patients who provided in-depth stroke decision-making data and indirectly in the network metrics. From the subgroup, we reported the first responder in Supplement Table 1, column 1. We learned from this group that the persons were usually strong ties for slow arrivers and weak ties for fast arrivers. The presence of strong ties versus weak ties is accounted for by the constraint and effective size metrics (as explained in Methods, P.14-15). Therefore, although not

explicitly identified, the strong tie or weak tie who could be available to the patient is indirectly captured in the network measures. We highlight this in the methods section.

P.15: “Notably, although we recorded the first responder only for the subgroup who provided qualitative data (discussed below), constraint and effective size account for the number of weak versus strong ties available to the patient during the emergency. Therefore, these are indirect measures of who could be present at the time of stroke.”

4. I did not find the network maps in Figure 2 to be particularly informative – it maybe best if this was in the supplementary information rather than the main text.

The network maps are the raw data that are important to show the data richness and the difference in the groups. We prefer to keep in the main text if possible. To aid in understanding the graphic, we have added a scale and median on the left margin. We have also added a figure legend at the end of the manuscript.

P.28: *Fig 2. Personal Networks of Patients with Ischemic Stroke, according to Slow and Fast Arrival.* Each black circle represents one patient in the study embedded inside his/her social network. Line color indicates strength of relationship: red lines are strong ties and blue lines are weak ties. Networks are arranged from highest constraint (top left) to lowest constraint (bottom right) with scale and median along the left margin. Slow arrivers’ networks were generally with higher constraint compared to fast arrivers’ networks.

Reviewer 2’s comments:

In this paper, the authors present a study that examines the relation between ego-network structure and delayed arrival at the hospital of 175 mild-stroke patients. The results show that small and close personal networks with strong ties -- after controlling for several confounders -- they are related with late arrival at the hospital.

There are definitely several limitations of the current study (including many that the authors acknowledged in the discussion). I would summarize the limitations as the inability to be confident that the study identifies causal evidence. On the other hand, the paper is very well written, the study technically sounds, and the topic is very interesting in both clinical and social research. For those reasons, I believe an improved version could be accepted for publication in Nature Communications. I have some remarks that need to be addressed:

- 1. I believe, if we exclude all other confounders, the fact that the authors do not consider as control if the patient was surrounded by “at least one ego-network member” at the time of the stroke is problematic. Do the authors have this information?**

We do not have the information on the whether a network member was present at the time of stroke. We did record the first responder for the subgroup of patients who provided in-depth stroke decision-making data. The availability of strong ties versus weak ties is accounted for by the network metrics, constraint and effective size, which we highlight in the methods section.

P.15: “Notably, although we recorded the first responder only for the subgroup who provided qualitative data (discussed below), constraint and effective size account for the number of weak versus strong ties available to the patient during the emergency. Therefore, these are indirect measures of who could be present at the time of stroke.”

2. I also believed that the 6-hour threshold was a bit arbitrary, until the authors addressed this at the robustness check section. My question is, why the authors did not consider the 4.5h threshold that a patient can get the tissue plasminogen activator.

We chose the 6-hour window, with 3-hour robustness check, because the 6-hour window has been related to favorable clinical outcomes in patients who did or did not receive reperfusion therapy (Matsuo et al., 2017). Not all mild stroke patients are candidates for treatment. Therefore, although earlier is better, the 6-hour threshold is the cutoff associated with favorable outcomes due to a combination of therapy, monitoring, and intervention if patients worsen. We describe this more thoroughly in the methods section:

P.13: “We chose the 6 hours threshold because it has been associated with favorable clinical outcomes, irrespective of whether patients receive reperfusion therapy or not.³ Particularly in mild stroke patients, who may not receive thrombolysis, this cutoff is associated with favorable outcomes due to a combination of early therapy, monitoring, and intervention if patients worsen.³”

3. The authors do not explain why they choose to work with a logistic regression. Why they do not consider the outcome to be just the continuous variable of delay time?

Despite our best efforts, we could not determine outside hospital timings for these patients with mild symptoms with enough precision and accuracy for a meaningful continuous outcome. A categorical outcome was much more accurate. We explain in the method section.

P.13: “We chose a categorical outcome, rather than continuous, because we could not determine the times with enough precision or accuracy for a meaningful continuous outcome.”

4. Table 2 results are little bit useless, since they are just unadjusted p-values of differences. I would suggest moving it to the supplementary materials – but I am not pushing on that comment.

In the spirit of showing our work, we felt that Table 2 shows the development process towards the full model. If this is too obvious, then we’re happy to move to supplementary, but we think it would be useful to the reader.

5. In the introduction there is no reference to networks effects, social capital/Burt’s concept, structural diversity/holes etc. There is a paragraph in the discussion, but I would suggest introducing these concepts in advance – especially for the audience that is not related to network science. Therefore, I am suggesting spending some lines on network effects (e.g. social influence in health behaviors – Christakis & Fowler, Strully et al., Aral & Nicolaides etc.) and then introduce the Burt’s concept (Burt 2000) and

strong and weak ties and structural diversity (Granovetter and thereafter work). As it is written, the authors refer directly in the results such concepts and may be difficult for the readers to follow.

We have added an introduction of these concepts, structured as advised by the reviewer, in the introduction.

P.3-4: The network perspective provides a set of theories and methods to study the spread of information or behaviors in groups. Research has shown that exercise,¹¹ weight gain,¹² and medication use¹³ may be contagious behaviors that flow through ties to shape and constrain personal choices. A formal way to study these influences is through well-known sociological theories known as the strength of weak ties¹⁴ and structural holes.¹⁵ These approaches illustrate two archetypal network formations of social capital: dense personal networks ideal for social support and radial personal networks optimal for access to novel information.¹⁶ Through these structural motifs, personal networks act as conduits of health-related social capital to identify symptoms, recognize a need for support, and help secure access to services,¹⁷ all of which are critical in medical emergencies.

6. Also, I would suggest If the authors want to understand the reasons why closed relatives prefer a ‘watch and wait’ pattern in relation to non-relatives, they could conduct surveys to the network of the patients rather than the patients themselves.

We agree. We added this as a future direction in the discussion.

P.12: A future direction of research could be to survey the network members after stroke or heart attack to understand why and how relatives more than non-relatives choose a watch and wait approach.

Reviewer 3’s comments:

1. The authors report that a small, closed social network structure is an independent predictor of delayed arrival after stroke onset. The study represents a fascinating new window into delayed arrival, the most important barrier to disease-changing stroke treatment, and will require acknowledgment in future studies of the topic. Strengths include clear expression of sociological concepts, sophisticated statistical analysis and inclusion of a semi-structured interview of a subgroup of patients to explore decision-making at stroke onset. However, the small study population (n=175) accumulated in two academic hospitals over 3 years, differed from the general stroke population in several important ways, some of which may be related to delayed arrival. It excluded patients who did not speak English or otherwise lacked capacity to participate in the survey interview, resulting in a very low average NIHSS of 3. Although low NIHSS is a powerful predictor of late arrival, only 38% arrived more than 6 hours after onset and 31% received IV tPA, a much higher rate than seen in the overall stroke population. Furthermore, the population was very young (average age 61) and highly educated (average 14 years of schooling). The authors appropriately identify the non-representative nature of the cohort as an important limitation but highlight exclusion of severe strokes when the NIHSS cutoff for the study was 19, far higher than the average.

Incapacity to participate in the questionnaire was likely a more important exclusion and might be highlighted instead.

As suggested, we have highlighted the incapacity to participate in the questionnaire as an exclusion.

P.12: However, exclusion of patients who could not participate in the interview survey, non-English speakers, patients with severe strokes or aphasia, patients who left before recruitment, and small sample size of slow arrivers may have introduced selection bias.

2. More broadly, would the authors consider characterizing the study throughout the paper as one of "mild stroke," not just "stroke." The impact of social networks likely diminishes when symptoms are more severe and this study has no capacity to assess that relationship.

Although we studied patients with mostly mild stroke (NIH stroke scale < 6, 88% of all participants), the cohort did include patients with moderate stroke (NIH stroke scale 6-15, 12% of all participants). After clarifying this point in the results section, we don't think it's accurate to state mild stroke throughout the manuscript. We stated "mostly mild" in relevant sections throughout the paper.

P.5: We recruited a diverse set of patients with mostly mild motor-predominant stroke (88% mild, 12% moderate). We focused on mild stroke because patients with milder symptoms are at higher risk of delay, and they were able to engage in the survey during hospitalization.

P.12: Our cohort include patients with mostly mild stroke.

P.13: The study was a cross-sectional design in which we recruited patients with mostly mild stroke from two academic hospitals, Barnes Jewish Hospital in St. Louis, MO and Brigham and Women's Hospital in Boston, MA, between May 2014 and May 2017.

REVIEWERS' COMMENTS:

Reviewer #1 (Remarks to the Author):

I thank the authors for addressing the issues I raised in my first review. They have all been dealt with satisfactorily within the confines of the data that they have available

Reviewer #2 (Remarks to the Author):

In this paper, the authors present a study that examines the relation between ego-network structure and delayed arrival at the hospital of 175 mild-stroke patients. The results show that small and close personal networks with strong ties -- after controlling for several confounders -- they are related with late arrival at the hospital.

There are definitely several limitations of the current study (including many that the authors acknowledged in the discussion). On the other hand, the paper is very well written, the study technically sounds, and the topic is very interesting to the broad scientific audience. In addition, following all reviewers' suggestions, the authors have made some substantial changes that enhanced the clarity of the analysis held.

For those reasons I believe that the paper can be accepted in the current form for publication in Nature Communications.

Reviewer #3 (Remarks to the Author):

The current use of the term "mild" in the manuscript doesn't fully address my concern. Here is a possible alternative to make the point more explicit:

"...Our results have limited generalizability. Our cohort include patients with mostly mild stroke and severity predicts time to arrival, so social networks may not be important in moderate or severe stroke."

Social Networks and Risk of Delayed Hospital Arrival after Acute Stroke

Response to the Referees' and Editor's Comments

Reviewer #3 comments:

- 1. The current use of the term "mild" in the manuscript doesn't fully address my concern. Here is a possible alternative to make the point more explicit: "...Our results have limited generalizability. Our cohort include patients with mostly mild stroke and severity predicts time to arrival, so social networks may not be important in moderate or severe stroke."**

We agree. We have incorporated this language, with some editing into 2 sentences, into the revised manuscript.

P.12: Additionally, our results have limited generalizability. Our cohort included patients with mostly mild stroke. Given that mild stroke is a risk factor for delayed arrival,⁴ we were examining a high-risk cohort. However, since severity predicts time to arrival, social networks may not be important in moderate or severe stroke.